# Exploration of Mangrove Endophytes as Novel Sources of Tannase Producing Fungi

**DOI:** 10.3390/jof11050366

**Published:** 2025-05-08

**Authors:** Vinodkumar Kushwaha, Jitendra R. Patil, Ganesh Chandrakant Nikalje, Lal Sahab Yadav

**Affiliations:** 1Post Graduate Research Laboratory, Department of Botany, Smt. Chandibai Himathmal Mansukhani College of Arts, Science & Commerce, Ulhasnagar 421003, Thane, MS, India; kushwahavinod30@gmail.com (V.K.); jrpatilrkt@gmail.com (J.R.P.); 2Department of Botany, Seva Sadan’s Ramchand Kimatram Talreja College of Arts, Science & Commerce, Ulhasnagar 421003, Thane, MS, India

**Keywords:** mangrove, endophytic fungi, screening, tannase

## Abstract

Tannase, a highly adaptive biocatalyst, plays a pivotal role in diverse bioconversion reactions in nature. This enzyme exhibits numerous applications across various industrial sectors, including food, pharmaceuticals, chemicals, and beverages. This study aimed to screen and characterize fungal endophytes isolated from mangrove plants for their enzyme tannase-producing ability. Eighty-five filamentous endophytic fungi were isolated from different mangrove samples and subsequently identified. These fungal strains were initially screened using the tannic acid agar plate method. Out of the screened strains, 13 fungal isolates demonstrated tannase production ability. The quantitative estimation of extracellular tannase was performed using the submerged fermentation technique. Among the studied endophytes, eight isolates, namely LV_084 (21.21 IU/mL), LV_074 (15.41 IU/mL), LV_078 (6.98 IU/mL), LV_038 (6.97 IU/mL), LV_077 (6.32 IU/mL), LV_016 and LV_066 (6.37 IU/mL), and LV_060 (6.18 IU/mL) exhibited excellent tannase activity. Among these isolates, LV_084 *Phyllosticta capitalensis* and LV_074 *Aspergillus chevalieri* showed the highest enzyme-producing ability. These isolates were authenticated using ITS rDNA sequencing, followed by BLAST search and phylogenetic analysis. Furthermore, the physical and chemical conditions for the maximum enzyme production were optimized. This is the first report of enzyme tannase production by *Phyllosticta capitalensis* and *Aspergillus chevalieri*.

## 1. Introduction

Mangrove ecosystems, at the intersection of terrestrial and marine environments, are well-known for their unique biodiversity and resilience. The microbial diversity, including bacteria, fungi, and actinomycetes, is rich in mangrove ecosystems and plants. Fungal communities in the mangrove habitat are diverse and are essential in nutrient recycling. Endophytic fungi are recognized for their ability to produce a range of bioactive compounds and enzymes that significantly contribute to the host plant’s defense mechanisms and ecological balance [1,2,3]. The harsh environment of mangrove habitats induces the fungal physiology, producing several biochemicals, including enzymes, by endophytes. Research on screening endophytic fungi for their potential to produce enzymes is gaining attention in this biotechnological era. The enzymes cellulases, proteases, amylases, lipases, laccases, and chitinases are industrially important enzymes obtained from fungal sources [4,5,6,7]. Tannase is less explored among the reports of important hydrolytic enzymes from mangrove endophytic isolates. However, some soil fungi, such as *Aspergillus*, *Penicillium*, and *Paecilomyces*, are known to produce extracellular tannase. Tannase is particularly noteworthy because it breaks down tannins into gallic acid, a compound with numerous applications in pharmaceuticals, food processing, and environmental cleanup.

Tannins are naturally occurring polyphenolic compounds present in various plant species [8]. They possess strong antibacterial properties and can precipitate proteins by forming complexes with enzymes and other proteins [9]. The enzymatic degradation of tannins by tannase offers an environment-friendly alternative to chemical catalysts, highlighting the potential of microbial tannase in sustainable industrial applications. Several fungi, such as *Penicillium*, *Aspergillus*, *Fusarium*, *Rhizopus*, *Trichoderma*, *Candida*, and *Saccharomyces*, are commonly explored for tannase enzyme production [10,11]. Tannase-producing fungi, such as *Aspergillus* and *Penicillium*, have been extensively studied for their biotechnological potential [12,13,14]. However, exploring mangrove endophytes as sources of tannase-producing fungi presents a promising opportunity to discover novel strains with enhanced enzymatic activities.

Mangrove endophytes, which are adapted to the challenging conditions of mangrove environments, may exhibit unique metabolic capabilities that enable them to thrive in these ecosystems. Isolating and characterizing tannase-producing fungi from mangrove endophytes could offer valuable insights into their enzymatic diversity and potential applications. This study aimed to investigate the diversity of tannase-producing fungi within mangrove endophytes, evaluate their enzymatic properties, and characterize the enzyme tannase.

## 2. Materials and Methods

### 2.1. Study Sites

The conserved mangrove region of the Godrej Mangrove Forest, Vikhroli, India (21.62° N, 108.23° E) was selected for this study (Figure 1). This site has a mean annual temperature of 27.2 °C, an average precipitation of 242.2 cm (95.35 inches), and a soil salinity of 0.5–1.2%. The dominant mangrove species observed at the site are *Avicennia marina* Vierh., *Acanthus ilicifolius* L., *Suaeda maritima* (L.) Dumort., *Derris indica* (Lam.), *Salvodora persica* L., *Acanthus ilicifolius* L., *Rhizophora mucronata* Lam., and *Derris trifoliata* Lour.

### 2.2. Sample Collection

The samples such as fresh and healthy leaves, twigs, and pneumatophores of all the abovementioned plants were collected from the Godrej Mangrove Forest, Vikhroli, India, from 2021 to 2023. A total of 40 samples were collected in three different seasons from selected plants and brought to the P.G. Research Laboratory, Department of Botany of the Smt. C.H.M. College, Ulhasnagar, Maharashtra, India. The samples were stored in a refrigerator at 4 °C until further isolation of endophytic fungi.

### 2.3. Isolation of Endophytic Fungi

The samples were washed with tap water to remove the dust and dirt from the surface and surface-sterilized using 1% sodium hypochlorite (NaOCl) and 75% ethanol solution for one minute, followed by washing with sterilized distilled water to remove traces of the sterilizing agent [15]. The surface-sterilized leaves, stems, and pneumatophores were cut into small segments (0.5 cm). Five segments of each plant part were aseptically placed onto the potato dextrose agar (PDA), potato carrot agar (PCA), Czapek–Dox agar (CzDA), and malt extract agar (MEA) media. Chloramphenicol (50 mg/L, HiMedia, Mumbai, India) was added to the media to suppress bacterial growth. All the plates were incubated at 30 °C for 10 days, and periodic observations were undertaken for the emergence of hyphae from the seeded segments. A small piece of growing hyphal tip from the segments was placed on fresh PDA plates for further growth and purification.

### 2.4. Morphological Characterization and Identification of Isolates

The pure isolates were transferred to PDA plates for colony characterization and growth study. A small portion of 5-day-old mycelia was aseptically placed on slides mounted with lactophenol cotton blue stain and sterile distilled water, then observed under a light microscope (Magnus MX21i-LED, Magnus Opto Systems India Pvt. Ltd., Noida, India). Images of fungal structures were captured using a MagcamHDL5MP camera, Magnus Opto Systems India Pvt. Ltd., Noida, India. The isolates were identified morpho-taxonomically using standard books and monographs [16,17,18,19,20,21,22,23,24,25,26]. All the identified fungal strains were deposited at the National Fungal Culture Collection of India, Agharkar Research Institute, Pune, Maharashtra, India.

### 2.5. Screening of the Isolates for Tannase Production

#### 2.5.1. Qualitative Screening (Plate Method)

The primary screening of tannase-producing isolates was performed on tannin agar media as per Pinto et al. [27]. The freshly grown 5-day-old fungal mycelia were centrally inoculated on tannin acid agar (TAA) media: peptone (5 g/L), sodium chloride (5.0 g/L), beef extract (1.5 g/L), yeast extract (1.5 g/L), tannic acid (5 g/L), and agar (20 g/L). The plates were incubated at 30 °C for 5–7 days. The isolates capable of utilizing tannic acid exhibited growth on TAA media. A dark zone around the colony indicated hydrolysis of tannins by isolates, which is considered a positive result. The zone diameters around the fungal colonies were measured at 72 h and 120 h, and the enzymatic index (EI) was calculated using the following formula:$$Enzymatic\;Index\;(EI)=\frac{Diameter\;of\;the\;clear\;zone-Diameter\;of\;the\;colony}{Diameter\;of\;the\;colony}$$

#### 2.5.2. Quantitative Screening (Liquid Broth)

The positive isolates were further screened for tannase production in a liquid broth. The isolates were inoculated in separate flasks containing 100 mL of liquid broth media containing peptone (5.0 g/L), sodium chloride (5.0 g/L), beef extract (1.5 g/L), yeast extract (1.5 g/L), and 1% tannic acid as the sole carbon source (pH 5.5). Five-day-old pure cultures of selected isolates were inoculated (5 mycelial plugs of 0.5 cm) in separate flasks containing sterilized production media. The inoculated flasks were incubated at 35 °C for 96 h at 150 rpm on a rotary shaker. After the incubation period, cultures were harvested, and the mycelial biomass was filtered using Whatman No. 1 filter paper. The free cell filtrate was used as a source of extracellular tannase and taken to determine tannase activity.

### 2.6. Enzyme Tannase Assay

The enzyme assay was based on the formation of the chromogen due to the reaction between gallic acid and rhodamine as described by Sharma et al. [28]. Enzyme tannase from the crude filtrate was quantified spectrophotometrically at 520 nm using rhodamine. A gallic acid standard curve was prepared by varying concentrations ranging from 0.125 mM to 4 mM. Methyl gallate was used as a substrate. One unit (U) of enzyme activity was determined as the amount of enzyme required to release one µmol of gallic acid per minute under standard assay conditions. The specific activity of the enzyme was calculated by dividing the enzyme activity by the protein concentration.

### 2.7. Protein Assay

The protein concentration of the samples (crude extract) was measured using Bradford’s method, with bovine serum albumin (BSA) as the standard [29]. The specific activity of the enzyme tannase was expressed as enzyme activity (IU) per milligram of protein (mg) in each sample.

### 2.8. Optimization of the Production of Extracellular Tannase Under SmF

Based on quantitative screening, hyper-tannase-producing isolates were cultured using submerged fermentation (SmF). The parameters such as incubation period, temperature, and pH were optimized for higher yields of a stable enzyme using the univariate parameter optimization method [30]. All the isolates were inoculated separately in 100 mL of production media (tannic acid broth with 1% w/v tannic acid) and incubated on a rotary shaker at 120 rpm.

#### 2.8.1. Incubation Period

A loopful culture of selected isolates was separately inoculated in flasks containing 100 mL of production media with a pH of 5.5. The inoculated flasks were incubated at 35 °C. The tannase production was estimated by periodically withdrawing 5 mL of the fermentation broth at 48 h, 72 h, 96 h, and 120 h. This experiment was performed in triplicate.

#### 2.8.2. Temperature

To determine the optimal temperature for enzyme production, sets of inoculated flasks with 100 mL of production media (pH 5.5) were incubated at different temperatures, 30 °C, 35 °C, 37 °C, and 40 °C, for the optimum incubation period, i.e., 96 h.

#### 2.8.3. pH

To determine the optimal pH, all the selected isolates were inoculated in production media with different pH levels (3.5, 4.5, 5.5, 6.5, and 7.5) and incubated at the optimum temperature (i.e., 35 °C) and incubation period (96 h).

### 2.9. Determination of Fungal Biomass

The fresh (wet) weight of fungal biomass was measured after 120 h of incubation at 35 °C in production media. At the end of the incubation time, mycelia were filtered through a circular Whatman No. 1 filter paper and weighed using a digital balance.

### 2.10. Molecular Identification of LV_074 and LV_084

The hyper-enzyme-producing isolates were authenticated using molecular methods. Five-day-old cultures grown on PDA media at 30 °C were used for DNA extraction. DNA was extracted using the CTAB method [31]. The targeted DNA regions were amplified using the polymerase chain reaction (PCR) by outsourcing the samples to Barcode Bioscience Pvt. Ltd., Bangalore, India. The ITS region was amplified using the primers ITS4/ITS5 [32,33]. The amplified products were purified using a Qiagen (Cat. No. 28104) PCR kit, QIAGEN Pvt. Ltd., Germantown, MD, USA and sent for sequencing to Barcode Bioscience Pvt. Ltd., Bangalore, India. The sequences of the isolates were compared with the NCBI GenBank database using BLASTn (https://blast.ncbi.nlm.nih.gov/Blast.cgi?PROGRAM=blastn&PAGE_TYPE=BlastSearch&LINK_LOC=blasthome, accessed on 20 March 2025) to identify the closest matching sequences.

### 2.11. Sequence Alignment and Phylogenetic Analysis

ITS sequences of all the closely matched sequences were downloaded and assembled using sequences of all the available species described in the database. The datasets were aligned using MAFFT v.7 (https://mafft.cbrc.jp/alignment/server; accessed on 25 March 2025) [34]. The sequences were manually edited using MEGA version 7 [35]. Maximum likelihood (ML) analysis was performed with IQTree v2.4.0 [36]. The phylogenetic trees were viewed and arranged using Interactive Tree of Life (iTOL) v4 (https://itol.embl.de/; accessed on 25 March 2025) [37]. The newly obtained sequences were deposited in the GenBank database. 

### 2.12. Statistical Analysis

Assays were performed in triplicates, and the values were expressed as the means ± standard deviations using MS Excel 2019.

## 3. Results

### 3.1. Isolation of Tannase-Producing Fungi

In this study, 85 endophytic fungi were isolated from mangrove plant species from the Godrej Mangrove Forest, Vikhroli, India. Most of the endophytic fungal strains were recovered from *A. marina* and *A. officinale* which covered 31.6% and 14.1% of the total isolates. These two plants are true mangrove plants and more dominant in the study area (Figure 2). This region has a lower abundance of *Rhizophora mucronata*, *Salvadora persica*, *Derris indica*, and *Suaeda maritima*. However, a good number of endophytic fungi were also obtained from different parts of the selected plants. Based on morphological and microscopic observations, 25 genera and 56 species from the different classes of fungi were identified. The predominant genera were *Chaetomium*, *Aspergillus*, *Scopulariopsis*, *Microascus*, *Alternaria*, and *Fusarium*. Some isolates were unable to produce reproductive structures even after a long period of incubation on different culture media. To identify such isolates morphologically, clamp connections were seen in the mycelium. Therefore, they were considered as basidiomycete isolates. All 85 isolates were screened for their tannase-producing ability. Among them, 13 active isolates showed a positive result (Table 1).

The morphological study on all tannase-producing fungi was carried out on PDA media. The selected isolates were morphologically distinct from each other. Based on morphological characters, the isolates were identified, and the authenticity of identification was confirmed by the Fungal Identification Service, Agharkar Research Institute, Pune. GenBank accession numbers are listed in Table 2. In this study, eight different genera were isolated from different mangrove plants, including dematiaceous fungi; most isolates belonged to ascomycetes and anamorphic ascomycetes. Morphological studies enabled the identification of dematiaceous fungi *Curvularia brachyspora*, *Curvularia lunata*, *Alternaria tenuissima*, *Corynespora cassiicola*, and *Cladosporium* sp. based on colony color, conidiophores, and conidial morphology. In contrast, the ascomycetous genus *Chaetomium* was determined based on ascomata, asci, and ascospores. The genera such as *Aspergillus*, *Penicillium*, and *Phyllosticta* are very complex, with more species; hence, they were initially determined at the genus level based on colony morphology and conidial characters. Furthermore, the fungi that showed hyperactivity of the enzyme tannase were authenticated using ITS rDNA sequence analysis. The authenticated fungi were deposited for accession number in the National Fungal Culture Collection of India, ARI, Pune.

### 3.2. Primary Screening of Tannase-Producing Fungi

Tannic acid agar (TAA) plates were used for primary screenings of isolates for their tannase-producing ability as described by Bradoo et al. [38]. Approximately 13 isolates could grow on TAA medium containing 1% tannic acid as the carbon source. After 72 and 120 h of incubation, the tannase hydrolysis activity in a dark zone around the colony was observed. The diameter of the hydrolysis zone was measured, and the enzymatic index was calculated (Table 1). The photographs showing the dark zone around the colony produced by isolates are shown in Figure 3. With the increase in time, the isolates’ colony diameter and zone diameter increased, confirming the utilization of tannic acid as the carbon source. The enzymatic activity of the isolates differed significantly in terms of the enzymatic index (Table 1). The enzymatic index after 72 h of incubation ranged from 0.17 to 2.28; however, after 120 h, it was not possible to calculate the enzymatic index for isolates LV_002, LV_022, LV_038, LV_060, and LV_074 due to mycelial growth covering the entire plate.

The enzymatic index of some isolates could only be measured after 120 h of incubation due to a slow growth rate. After 120 h of incubation, four isolates exhibited an enzyme index greater than 1.0. Among them, isolates LV_047 and LV_084 showed the highest indices of 1.54 and 1.45, while isolates LV_002 and LV_074 demonstrated the maximum indices of 2.28 and 2.1 after 72 h of incubation and were characterized by very rapid growth.

The plate assay is considered an easy and quick screening method that indicates the microbial capacity to use tannic acid as the carbon source [27]. Although 13 fungal strains could thrive on the TAA medium containing 1% tannic acid, the possibility of producing maximum enzymes under submerged fermentation for extracellular tannase was to be examined. All the positive isolates were further analyzed for extracellular enzyme tannase production in a liquid broth.

### 3.3. Quantitative Estimation of Tannase Under SmF

After inoculating all 13 isolates individually in a liquid broth and incubating on a rotary shaker for 96 h, extracellular tannase assays were carried out. A total of 4 isolates exhibited excellent tannase activity in the liquid (Figure 4); among them, isolates LV_074 and LV_084 were hyper tannase producers. The enzyme tannase activity results showed that fungal isolates LV_084 and LV_074 isolated from *Avicennia marina* showed the highest tannase activity with 21.21 IU/mL and 15.41 IU/mL at 96 h, followed by other fungal isolates LV_038 (6.97 IU/mL) and LV_016 (6.37 IU/mL).

The highest tannase activity for all the fungal strains was observed after 96 h. The low tannase activity recorded after 48 h of incubation was because of less mycelial growth in the early stage in the production media; thus, less extracellular tannase was synthesized to break down the tannic acid present in the media. Due to their highest tannase activity, the LV_084 and LV_074 isolates were chosen as the best tannase-producing fungi. Furthermore, physicochemical parameters of tannase production by isolates LV_084 and LV_074 were performed in control conditions.

### 3.4. Optimization of Conditions for Tannase Activity of LV_074 and LV_084

#### 3.4.1. Optimization of the Incubation Period

The optimization of the incubation period for the maximum tannase activity was carried out at 48, 72, 96, and 120 h of incubation (Figure 5). The results revealed that from 48 h to 96 h, both isolates showed a significant increase in tannase activity at 96 h (from 1.5 IU/mL to 15.48 IU/mL in LV_074 and from 2.5 IU/mL to 21.21 IU/mL in LV_084). Later, the tannase activity decreased significantly at 120 h (11.5 IU/mL in LV_074 and 15.96 IU/mL in LV_084). The optimum incubation period for the maximum tannase activity for both isolates was 96 h.

#### 3.4.2. Optimization of the Incubation Temperature

Temperature is another important factor in the growth and metabolic activities of fungi, consequently in extracellular enzyme production. This work examined four different temperatures: 30, 35, 40, and 45 °C (Figure 6). The results showed that at 30 °C, the enzyme activity was lowest, while the highest activity was observed at 35 °C. Generally, the concentration of the enzymes gradually decreases at 40 °C and 45 °C.

#### 3.4.3. Optimization of the pH Level

The pH level of microbial growth media is one of the most important factors for microbial growth and enzyme production. However, the optimum pH for microbial growth is not necessarily the same as the optimum pH for enzyme activity. The pH level of the growth medium plays a vital role in enzyme secretion by microorganisms. This study applied five different pH levels: 3.5, 4.5, 5.5, 6.5, and 7.5. The tannase activity was low at 3.5 pH after 96 h of incubation (Figure 7) and increased in both isolates at pH 5.5 (15.48 IU/mL in LV_074 and 21.21 IU/mL in LV_084). However, after pH 5.5, it decreased. Thus, pH 5.5 was the optimum condition for the maximum tannase activity.

### 3.5. Biomass Production

The biomass dynamics of isolates during the extracellular secretion of tannase were illustrated in Figure 8. As time progressed, the biomass of the isolates increased, confirming the utilization of tannic acid as the carbon source. The biomass of both isolates LV_074 and LV_084 increased with incubation period, and the maximum biomass was measured to be 4.292 mg and 3.492 mg, respectively, at 96 h. However, thereafter, there was no significant change in biomass.

### 3.6. Molecular Identification of the Maximum Tannase Producer: LV_074 and LV_084

The isolated strains LV_074 and LV_084 were identified as *Aspergillus chevalieri* and *Phyllosticta capitalensis* using sequencing of the ITS-4 and ITS-5 of ribosomal DNA and then BLASTn. Furthermore, the phylogeny based on the ITS rDNA gene sequence confirmed the identity of isolates LV_074 and LV_084.
**LV_074. *Aspergillus chevalieri* var. *chevalieri* (L. Mangin) Thom & Church**


The BLAST (https://blast.ncbi.nlm.nih.gov/Blast.cgi?PROGRAM=blastn&PAGE_TYPE=BlastSearch&LINK_LOC=blasthome, accessed on 20 March 2025) search results on the ITS sequence of LV_074 showed 100% sequence similarity with *Aspergillus chevalieri* accession No. LN482478.1 and *Aspergillus montevidansis* accession No. OW987704.1. The phylogenetic relationship within the restriction section of the genus *Aspergillus* revealed that LV_074 forms a topology with *Aspergillus chevalieri,* so isolate LV_074 is assigned as *Aspergillus chevalieri* and taxonomically described (Figure 9), GenBank accession No. PV054919. The phylogram is given in Figure 10.

**Description.** Colonies: on PDA, sulfur yellow-to-orange-colored, velvety to slightly sulcate, slow-growing (15–25 mm in diam.), in 15 days, at 35 °C. Mycelium: wide, septate, granulate, shiny structure on the surface of the colony due to formation of ascomata, irregular margins, sporulation sparse to moderately dense, with green conidial mass. Conidiophores: short (70–120 × 4–6 μm), hyaline, and smooth-walled. Vesicles: pyriform, 20–48 μm in diameter. Phialides: ampulliform, 5.5–7.5(–10) × 3–5 μm, uniseriate, conidial head radiating with conidial mass. Conidia: 3–4(–6) × 2.5–3.5(–5) μm, globose to subglobose, hyaline, rough-walled. Ascomata: 100–250 μm in diameter, cleistothecial, superficial, yellow, globose to subglobose. Ascospores: globose to subglobose, smooth to slightly verruculose, 3.5–5.5 × 3–4 μm, lenticular furrow present, crests of 0.5–1 μm, smooth-walled to roughened.

**Habitat.** Isolated from mangrove plant *Avicennia marina*, Thane-Mumbra creek, Mumbai, Maharashtra India
Figure 9*Aspergillus chevalieri:* (**a**) growth on TAA showing a dark zone around the colony; (**b**) colony on PDA verse; (**c**) PDA reverse; (**d**) a conidiophore with a vesicle; (**e**) a vesicle with sterigmata; (**f**) conidia; (**g**) ascomata; and (**h**) ascospores.
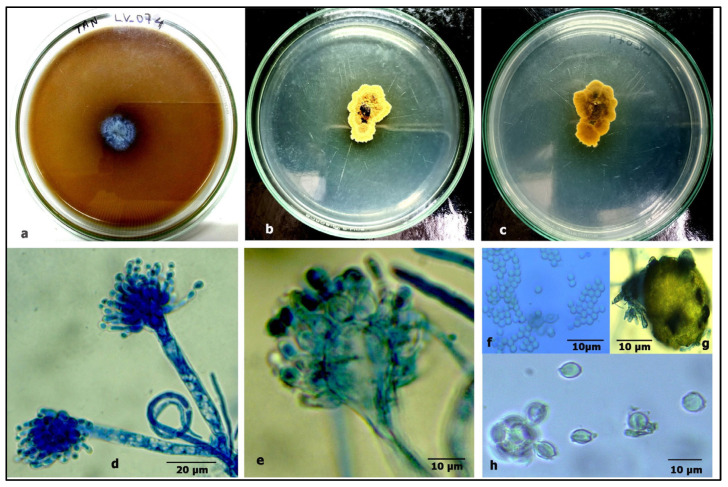

Figure 10Phylogram of *Aspergillus* resulting from the maximum likelihood (RAxML) tree using ITS sequences. Confidence values for ML ≥50% (UFboot2/RAxML) are included near the nodes. The specimens described in this study are highlighted in bold. *Aspergillus xerophilus* NRRL6131, *Aspergillus xerophilus* NRRL6132, and *Aspergillus osmophilus* IRAN2090C were used as an outgroup.
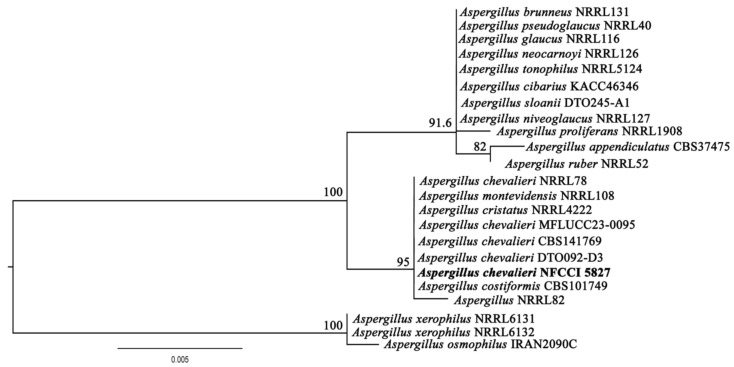

**LV_084. Phyllosticta capitalensis Henn.**

Based on the BLAST search and phylogenetic analysis of the ITS sequence, the isolate LV_084 is described as *Phyllosticta capitalensis* (Figure 11), GenBank accession No. PV054914. The phylogram is given in Figure 12.

**Description.** Colonies: on PDA, greenish-black to light grey, slow-growing (30–40 mm), in 15 days, at 35 °C. Reverse black, irregular in shape, with sparse mycelium. Mycelium: wide, branched, septate, light dematiaceous and globulate. Conidiomata: pycnidial type, black-to-grey-colored, round to elongated (up to 35 μm), pycnidial walls multilayered, dark brown with textura angularis. Ascomata: similar to conidiomata. Asci: clavate to broadly ellipsoid, bitunicate, hyaline, 45–85 × 9–13 μm. Ascospores: hyaline, aseptate, bi- to multi-seriate, unicellular, broader in the middle, with obtuse colorless ends with a terminal mucoid appendage, smooth-walled, 15–18 × 6–7 μm.

**Habitat**. Isolated from mangrove plant *Avicennia marina*, Kalher creek, Mumbai, Maharashtra, India.

## 4. Discussion

Fungal endophytes represent a significant component of fungal communities. Interest in endophytic fungi is predicated on several factors; they constitute the primary reservoir of biodiversity within the fungal kingdom [39], encompassing a diverse array of taxa from eumycota, predominantly ascomycetes and their coelomycetous anamorphs, along with the occasional occurrence of basidiomycetes. In the present study, it was found that ascomycetous fungi predominantly colonized mangrove plants. The ascomycete fungi were isolated from all the selected mangrove plants frequently. In anamorphic ascomycete fungi, dematiaceous fungi were dominant along with coelomycetous anamorphs. These results support the findings of Du et al. [40], Mishra et al. [41], and Pecoraro et al. [42]. They reported that ascomycetes were the most abundant members of endophytic fungal communities isolated using traditional separation techniques. However, it is essential to note that diversity analyses of endophytes based on culture-dependent methodologies underestimate actual diversity. Many isolates may still be identified using culture-independent methods [43].

The cultural characteristics of endophytic fungi were significantly different from the strains isolated from soil or any other habitat. The mycelial growth and reproductive structure formation by endophytic fungi were slow; some isolates exhibited reduced sporulation. Therefore, morphological identifications are somewhat tricky. Table 2 revealed that the tannase-producing isolates were from eight different genera; the predominant genera were *Chaetomium*, *Aspergillus*, and *Curvularia.* The fungal genus *Phyllosticta*, a plant pathogen, isolated from the healthy leaves of *A. mariana*, causes leaf spot disease.

Studying fungi and their role in biotechnological applications has proven significant in industry and sustainable agriculture. Several noteworthy strains of fungi are known for their beneficial secondary metabolites and ability to produce enzymes. Exploring potent and industrially important fungi from various habitats has been an ongoing global research. Mangrove endophytic fungi have attracted considerable attention in biotechnology due to their capacity to produce bioactive substances with diverse applications. Researchers have explored mangrove-associated fungi for enzymes such as amylases, cellulases, pectinases, proteases, and chitinases [4,5,6,7]. Enzymes extracted from these organisms have many commercial and industrial uses [44]. Due to its wide applications, tannase stands out among the array of enzymes as a potential game-changer in various industrial sectors. However, the mangrove ecosystem is the least exploited among the habitats explored for tannase-producing fungi.

In the present study, 13 isolates were obtained as potent tannase enzyme producers (Table 1 and Table 2). The most tannase-producing genera were identified as *Chaetomium*, *Aspergillus*, *Curvularia*, *Alternaria*, *Corynespora*, and *Phyllosticta.* The records of tannase producers from mangrove habitats are meagre and restricted to *Aspergillus niger*, *A. japonicas*, *A.aculeatus*, *A. fumigatous*, and *Penicillium* species [45,46]. To the best of our knowledge, this is the first report demonstrating the ability of the mangrove endophytic *Alternaria tenuissima*, *Curvularia brachyspora*, *Curvularia lunata*, *Corynespora cassiicola*, and *Phyllosticta capitalensis* to produce tannase, thereby expanding the microbial sources for this industrially significant enzyme from a unique habitat. The specific tannase enzyme production ability under the specified conditions by the isolates *Alternaria tenuissima* (2.38 IU/mL), *Curvularia brachyspora* (3.0 IU/mL), *Curvularia lunata* (3.81 IU/mL), *Corynespora cassiicola* (3.93 IU/mL), and *Phyllosticta capitalensis* (21.22 IU/mL), respectively, was very comparable to known tannase-producing isolates. The mangrove ecosystem is rich with tannin-containing plants. The soil contains more organic matter due to this specific feature of the ecosystem, which supports microbial growth. Microbes have the potential to degrade tannins as an energy source [45].

Several tannase-producing fungi have been reported; among them, *Aspergillus* sp. and *Penicillium* sp. are the most common filamentous fungi involved in tannin bioconversion, individually or in co-cultures [12,47]. Furthermore, various other fungi from the genera *Trichoderma*, *Fusarium*, *Chaetomium*, *Rhizoctonia*, *Candida*, and *Saccharomyces* have also been reported for their ability to degrade tannins, particularly hydrolyzable tannins [10,11,48,49]. However, the genus *Aspergillus* has been the most potent and extensively studied tannase producer among the existing fungal sources [13,14,50]. Batra and Saxena [12] screened 35 *Aspergillus* and 25 *Penicillium* isolates for their tannase-producing ability, and it was found that 25 *Aspergillus* and 20 *Penicillium* isolates exhibited tannase activity in solid as well as in a liquid broth. Furthermore, endophytic *Penicillium rolfsii* CCMB 714 isolated from cocoa (*Theobroma cacao*) leaves also showed high tannase production when using tannic acid as a substrate (9.4 U/mL) [51]. In this study, *Aspergillus* was the predominant genus with three different isolates: LV_010 *Aspergillus flavus* (2.522 IU/mL), LV_016 *Aspergillus* sp. (6.37 IU/mL), and *Aspergillus chevalieri* (15.41 IU/mL) exhibited significant tannase activity in a liquid broth. This finding validates the previous records reported by researchers. The genus *Aspergillus* is robust, able to tolerate extreme conditions, and known for several industrially applicable metabolites, enzymes, and organic acids. This may be due to its adaptability to thrive, genetic potential, and evolutionary pressure to produce diverse metabolites to survive. The genus *Chaetomium*, commonly considered for its vigorous cellulolytic activity, also predominantly colonized mangrove plants and exhibited good tannase activity in this study [4]. The isolate LV_038 *Chaetomium globosum* showed high tannase production (6.97 IU/mL).

The most potent tannase producers LV_074 and LV_084 were identified as *Aspergillus chevalieri* and *Phyllosticta capitalensis*, with GenBank accession numbers PV054919 and PV054914. The tannase activity of the above isolates was confirmed qualitatively and quantitatively. Further optimization of tannase production by LV_074 and LV_084, such as of the pH level, temperature, and duration, was studied in a liquid medium. The optimization conditions revealed that tannase synthesized by both isolates exhibited the highest gallic acid production at 35 °C and a pH of 5.5 (15.41 IU/mL and 21.21 IU/mL) after 96 h of incubation; they are best-suited for tannase production. Both isolates are mesophilic, thriving at temperatures between 25 °C and 38 °C; at 35 °C, the maximum growth was recorded. On the other hand, slightly acidic media (pH 5.5) supported optimal growth; hence, the above physicochemical conditions were very suitable for enzyme secretion. In agreement with this finding, the optimal activity of the purified tannase from endophytic *Aspergillus niger*, *Penicillium samsonii*, and *P. minioluteum* was obtained at 35 °C and pH 5.5 [52,53,54]. The incubation at 40 °C and pH 6.0 led to a reduction in the activity. These findings also align with endophytic *Aspergillus sydowii*, which showed an optimal pH range of 5.0 to 6.0. [55].

## 5. Conclusions

This study explored several tannase-producing endophytic fungal strains isolated from mangrove plants for the identification of novel sources of the tannase enzyme. These strains demonstrated significant tannase production when cultivated in submerged fermentation using 1% tannic acid as a substrate. The potential isolates *Aspergillus chevaleris* and *Phyllosticta capitalensis* exhibited the highest stable tannase-producing ability across a wide range of pH and temperature conditions, highlighting their potential for biotechnological applications. In addition, other isolates such as *Alternaria tenuissima*, *Curvularia brachyspora*, *Curvularia lunata*, and *Corynespora cassiicola* were reported for the first time as tannase producers. However, the optimization of physicochemical parameters for the maximum production of a stable tannase enzyme from these isolates must be standardized. The exploration of tannase-producing organisms is crucial due to the growing demand for cost-effective and stable enzymes derived from untapped biological sources. This study provides a pioneering report on tannase-producing endophytic isolates *A. chevaleris* and *P. capitalensis* from the mangrove plant *Avicennia marina*, contributing valuable insights into the potential of these fungi for future applications in enzyme production. More research studies are required on optimizing large-scale, cost-effective production of tannase and exploring marker genes involved in the biosynthesis of tannase.

## Figures and Tables

**Figure 1 jof-11-00366-f001:**
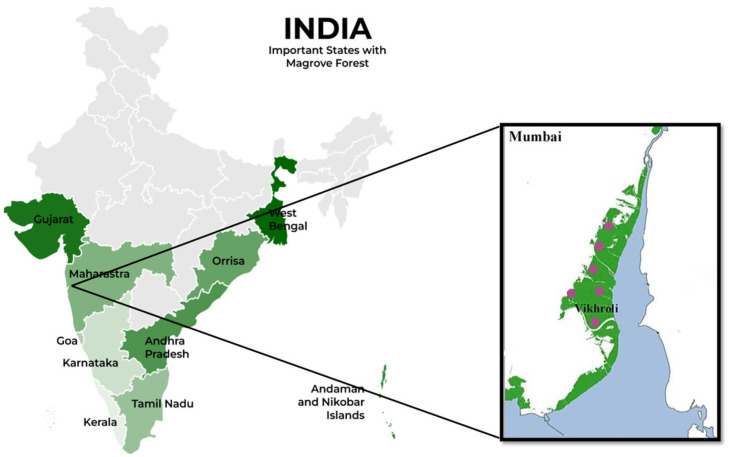
The map represents the mangrove ecosystem in different states of India . The green zone in the box represents Mumbai coastal region. Pink coloured spots represent sample collection sites in the Godrej Mangrove Forest at Vikhroli.

**Figure 2 jof-11-00366-f002:**
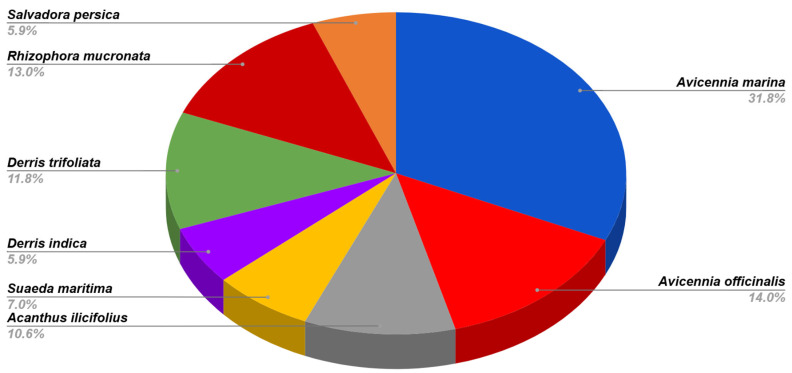
Diversity of fungal endophytes in *A. marina*, *A. officinalis*, *Acanthus ilicifolius*, *S. maritima*, *D. indica*, *D. trifoliata*, *R. mucronate*, and *Salvadora persica*.

**Figure 3 jof-11-00366-f003:**
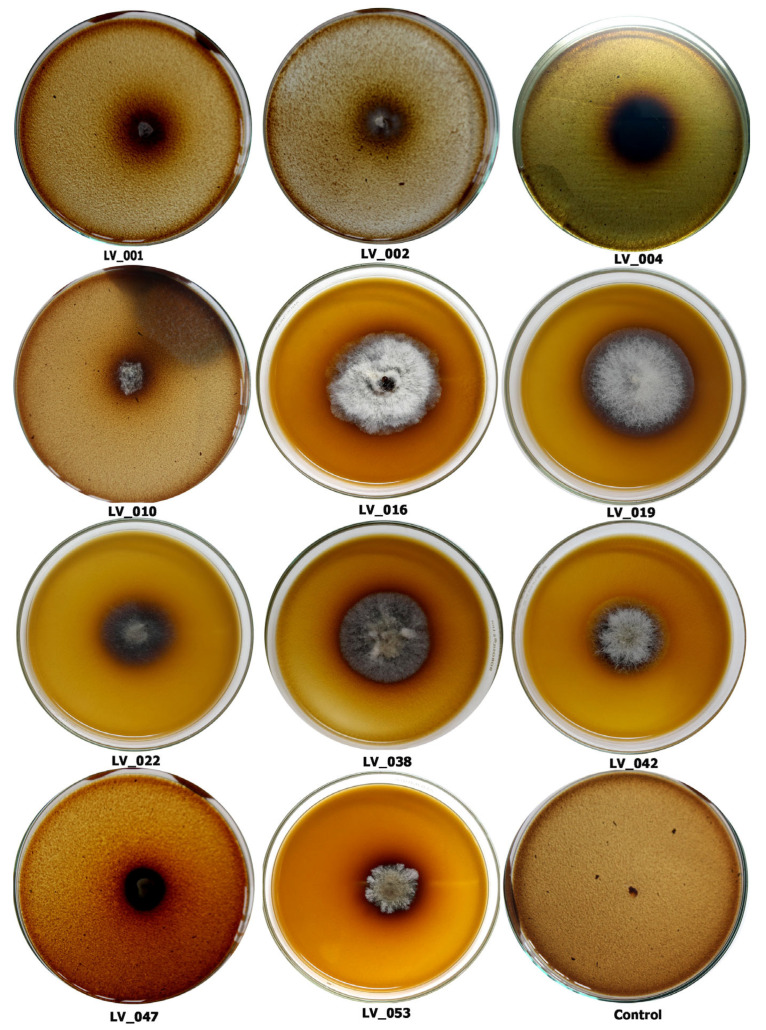
Fungi showing tannic acid degradation leading to the formation of a hydrolysis zone after 72 h: *Penicillium* sp. (LV_001); *Chaetomium* sp. (LV_002); *Chaetomium* sp. (LV_004); *Aspergillus* sp. (LV_010); *Aspergillus* sp. (LV_016)*; Curvularia brachyspora* (LV_019); *Alternaria tenuissima* (LV_022); *Chaetomium* sp. (LV_038); *Corynespora cassiicola* (LV_042); *Cladosporium limoniforme* (LV_047); *Curvularia lunata* (LV_053); and control.

**Figure 4 jof-11-00366-f004:**
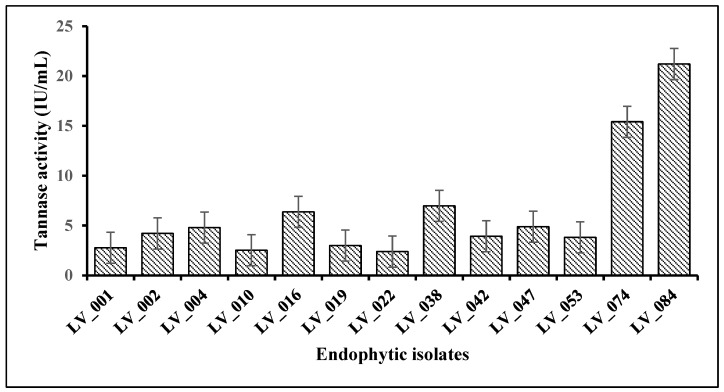
Quantitative estimation of the tannase enzyme activity of fungal isolates under submerged fermentation.

**Figure 5 jof-11-00366-f005:**
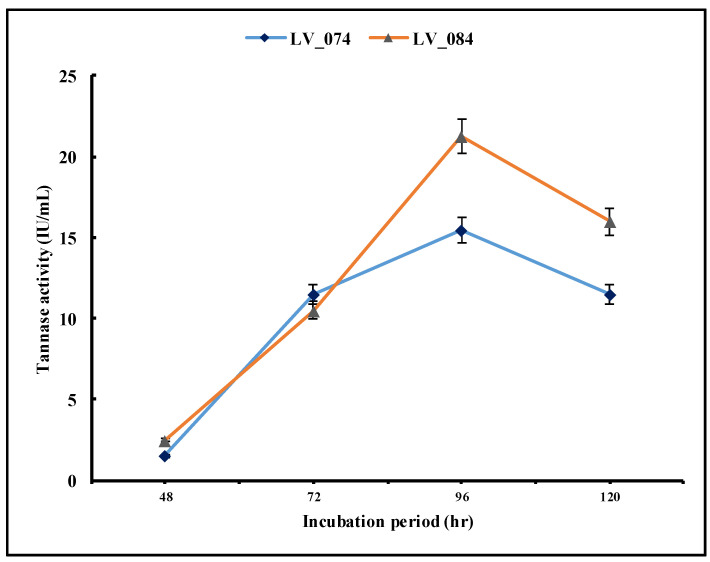
Optimization of the incubation period (48 h, 72 h, 96 h, and 120 h) for the maximum tannase activity of LV_074 and LV_084.

**Figure 6 jof-11-00366-f006:**
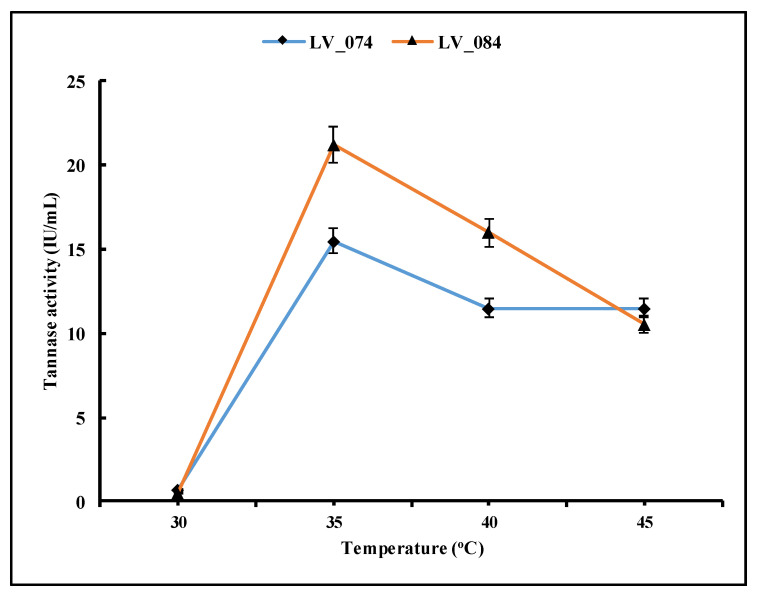
Optimization of the optimum temperature for the maximum tannase activity of LV_074 and LV_084.

**Figure 7 jof-11-00366-f007:**
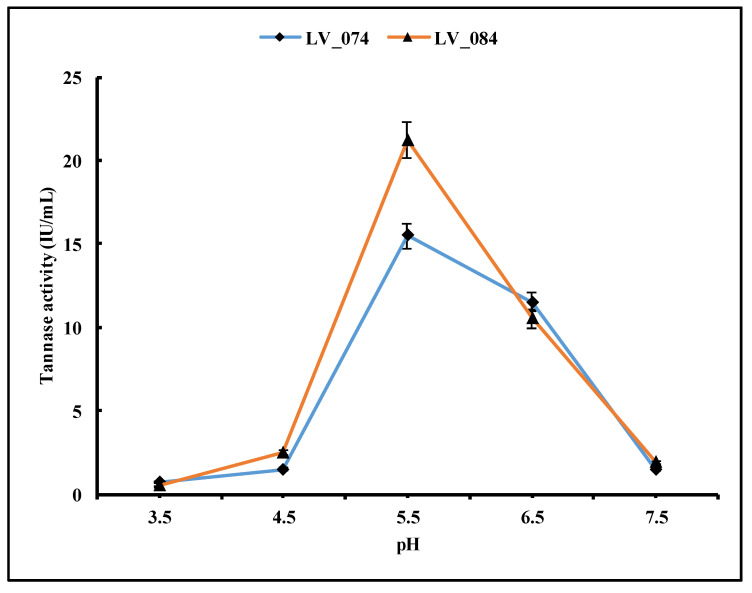
Optimization of the pH level of the growth medium for the maximum tannase activity in LV_074 and LV_084.

**Figure 8 jof-11-00366-f008:**
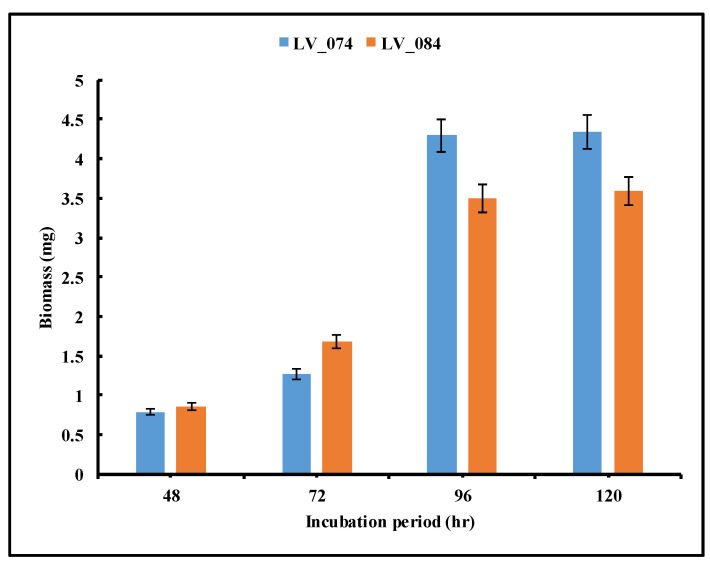
Optimization of the incubation period for the maximum biomass production of isolates LV_074 and LV_084.

**Figure 11 jof-11-00366-f011:**
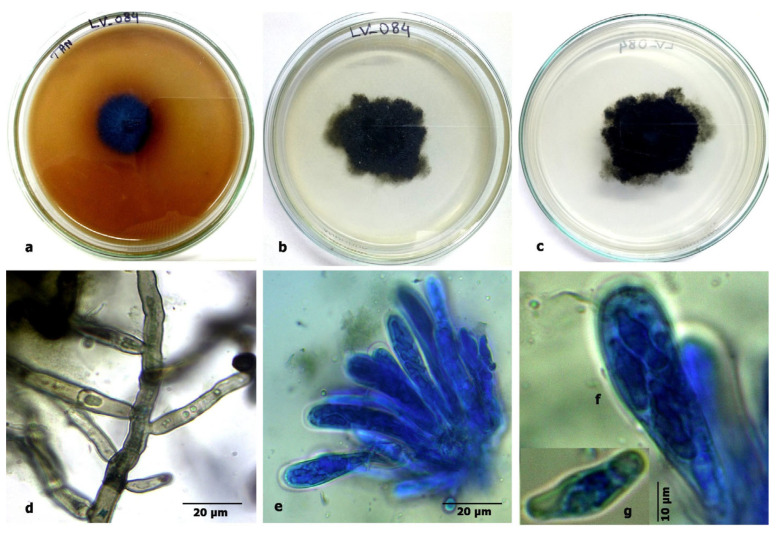
*Phyllosticta capitalensis* Henn.: (**a**) growth on tannic acid agar showing a dark zone around the colony; (**b**) colony on PDA verse; (**c**) PDA reverse; (**d**) mycelium; (**e**) an ascus; (**f**,**g**) asci and ascospores.

**Figure 12 jof-11-00366-f012:**
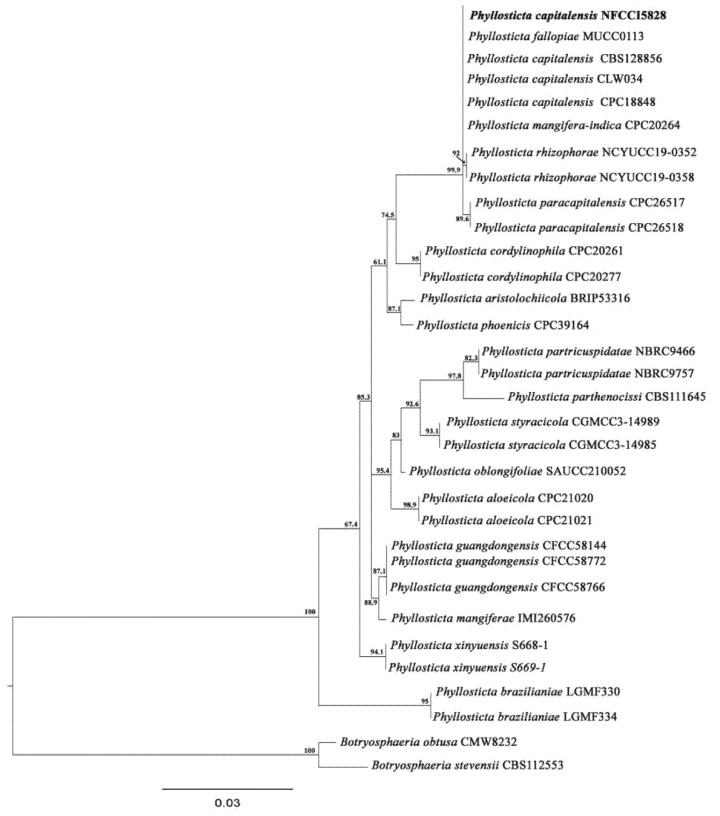
Phylogram of *Phyllosticta* resulting from the maximum likelihood (RAxML) tree using ITS sequences. Confidence values for ML ≥50% (UFboot2/RAxML) are included near the nodes. The specimens described in this study are highlighted in bold. *B. obtusa* (CMW 8232) and *B. stevensii* (CBS112553) were used as an outgroup.

**Table 1 jof-11-00366-t001:** Tannase-producing isolates and their tannin-hydrolyzing ability (enzymatic index) in plate assay.

Sr. No	Isolate No.	1% Tannic Acid Medium	Enzymatic Index After 72 h	Enzymatic Index After 120 h
1	LV_001	+	0.48	1.12
2	LV_002	+	2.28	*****
3	LV_004	+	0.57	0.78
4	LV_010	+	0.34	0.66
5	LV_019	+	0.73	1.34
6	LV_022	+	1.18	*
7	LV_038	+	1.20	*
8	LV_042	+	0.68	1.10
9	LV_047	+	0.91	1.54
10	LV_053	+	0.12	0.35
11	LV_016	+	0.41	*
12	LV_074	+	2.10	*
13	LV_084	+	0.30	1.45

‘+’ Positive result. * Unchecked—as the mycelial growth covered the entire plate.

**Table 2 jof-11-00366-t002:** List of tannase producing-isolates.

Sr. No.	Isolate Code	Host Plants	Identified Fungal Species	Accession No.
1	LV_001	*Avicennia marina* Vierh.	*Penicillium* sp.	NFCCI 5608
2	LV_002	*Avicennia marina* Vierh.	*Chaetomium* sp.	NFCCI 5505
3	LV_004	*Avicennia marina* Vierh.	*Chaetomium* sp.	NFCCI 5507
4	LV_010	*Acanthus ilicifolius* L.	*Aspergillus flavus gr.*	NFCCI 5508
5	LV_019	*Suaeda maritima* (L.) Dumort.	*Curvularia brachyspora* Boedijn	NFCCI 5510
6	LV_022	*Suaeda maritima* (L.) Dumort.	*Alternaria tenuissima* (Kunze) Wiltshire	NFCCI 5511
7	LV_038	*Avicennia marina* Vierh.	*Chaetomium globosum* var. *globosum* Kunze	CHMCC012
8	LV_042	*Acanthus ilicifolius* L.	*Corynespora cassiicola* (Berk. & M.A. Curtis) C.T. Wei	NFCCI 5513
9	LV_047	*Salvodora persica* L.	*Cladosporium* sp.	NFCCI 5616
10	LV_053	*Avicennia marina* Vierh.	*Curvularia lunata* (Wakker) Boedijn	NFCCI 5516
11	LV_016	*Derris trifoliata* Lour.	*Aspergillus* sp.	CHMCC003
12	LV_074	*Avicennia marina* Vierh.	*Aspergillus chevalieri* var. *chevalieri* (L. Mangin) Thom & Church	NFCCI 5827
13	LV_084	*Avicennia marina* Vierh.	*Phyllosticta capitalensis* Henn.	NFCCI 5828

## Data Availability

The original contributions presented in this study are included in the article. Further inquiries can be directed to the corresponding authors.

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
