# Peer review of "Exploration of Mangrove Endophytes as Novel Sources of Tannase Producing Fungi"

_jof, 2025, doi:10.3390/jof11050366_

Round 1
Reviewer 1 Report
This study offers a noteworthy advancement in the search for novel tannase-producing microorganisms, particularly from mangrove-associated endophytic fungi. The identification of Aspergillus chevalieri and Phylostricta capitalensis as promising producers of stable tannase enzymes across varied pH and temperature conditions is commendable. The potential biotechnological applications are well-stated, and the work addresses an important gap in enzyme sourcing from natural environments. To further enhance the quality of the manuscript, it is essential to revise the language for improved clarity and fluency (a thorough English language check across the entire article is necessary). Additionally, expanding the discussion on the possible mechanisms contributing to enzyme stability and including suggestions for future work, such as enzyme purification, would add greater depth and impact to the study.
Introduction
Please expand the Introduction to include more detailed information on the metabolic capabilities of mangrove endophytes, as this aspect is currently underexplained and lacks sufficient context.
Line 46: Reference 5 is outdated; please replace it with more recent studies to ensure the information reflects current research. Additionally, if you claim that the topic has been extensively studied, support this statement with multiple relevant references rather than relying on a single source.
Line 60 – 63: Please remove any mention of study results from the Introduction section. Only the objective or aim of the study should be stated at the end of the Introduction, without presenting findings or outcomes.
Materials and Methods
Line 77: In what year were samples collected?
Line 96: Some?
Line 184: Please add which statistical analysis was performed and what software was used.
Results
Line 193: replace „are“ with „is“
Line 195: delete „were“
Line 207: delete „were“
Please improve Figures 5 – 7 for better visual clarity.
Discussion
Lines 392 – 394: Improve english in whole sentence, it is not clear what the authors want to say
Line 419: delete „by“ and line 422 delete „were“
Line 423: Why 35°C and pH 5.5 are optimal conditions for both isolates? Is this consistent with known enzymatic properties of microbial tannases? Could strain-specific adaptations be discussed?
Lines 426 – 428: The comparison with other studies (references [45–48]) is valuable. However, these references should be better integrated into the discussion. Provide more details about how closely related these strains are and whether the same substrates or culture conditions were used.
Conclusions
Consider adding some directions for future studies.
Author Response
Reviewer#1:
Comments:
Comment: Please expand the Introduction to include more detailed information on the metabolic capabilities of mangrove endophytes, as this aspect is currently underexplained and lacks sufficient context.
Response: Thanks for the suggestions. We have incorporated a paragraph on the suggested point in the introduction section.
Comment: Line 46: Reference 5 is outdated; please replace it with more recent studies to ensure the information reflects current research. Additionally, if you claim the topic has been extensively studied, support this statement with multiple relevant references rather than relying on a single source.
Response: Corrected, three recent references have been included in the text.
Comment: Line 60 – 63: Please remove any mention of study results from the Introduction section. Only the objective or aim of the study should be stated at the end of the Introduction, without presenting findings or outcomes.
Response: Thanks for this suggestion. We have removed the entire paragraph.
Materials and Methods
Comment: Line 77: In what year were samples collected?
Response: The samples were collected from 2021 to 2023.
Comment: Line 96: Some?
Response: To make a slide for microscopic study, a small amount of mycelia from the margins of the growing colony was taken and mounted on the slide.
Comment: Line 184: Please add which statistical analysis was performed and what software was used.
Response: We used SPSS software to calculate standard deviation and standard error.
Results
Comment: Line 193: replace „are“ with „is“
Response: Corrected.
Comment: Line 195: delete „were“
Response: Corrected.
Comment: Line 207: delete „were“
Response: Corrected.
Comment: Please improve Figures 5 – 7 for better visual clarity
Response: Thanks for the suggestion. We have improved the clarity of figures( 600 DPI)
Discussion
Comment: Lines 392 – 394: Improve english in whole sentence, it is not clear what the authors want to say
Response: Many thanks. Changed the sentence
- Line 419: delete „by“ and line 422 delete „were“
Response: corrected.
Comment: Line 423: Why 35°C and pH 5.5 are optimal conditions for both isolates? Is this consistent with the known enzymatic properties of microbial tannases? Could strain-specific adaptations be discussed?
Response: Thanks for the comments. Both fungi are mesophilic, thriving at temperatures between 25 and 38 °C. At 35°C, maximum growth was recorded. On the other hand, the slightly acidic media pH (5.5) supports optimal growth; hence, the above physico-chemical conditions were very suitable for enzyme secretion. As per published data on tannase-producing fungi, most of the fungal isolates exhibited maximum tannase within the range of 30-35°C. Enzyme tannase is a temperature-sensitive enzyme that tends to maintain a stable structure, enabling efficient catalytic activity.
Comment: Could strain-specific adaptations be discussed?
Response: Many thanks for the suggestion. We have mentioned in the text.
Comment: Lines 426 – 428: The comparison with other studies (references [45–48]) is valuable. However, these references should be better integrated into the discussion. Provide more details about how closely related these strains are and whether the same substrates or culture conditions were used.
Response: Thanks for the suggestion. We have included and compared more findings; however, there are very few reports on tannase-producing mangrove endophytes.
Conclusions
Comment:. Consider adding some directions for future studies
Response: Many thanks for the suggestion. We have added a future plan to the text.
Reviewer 2 Report
The manuscript submitted for consideration is devoted to the search for promising producers of tannase, which is widely used in various fields. The article has a screening direction, which is an important biotechnological approach to identify promising species. The authors have managed to obtain convincing results and new data for science that could potentially have applied significance. An important aspect is to establish optimal parameters for the growth of selected isolates and tannase production.
It is advisable to indicate the authors of Host plants (lines 68-71) at the first mention of the genus and species of the plant in accordance with the International Code of Nomenclature for algae, fungi, and plants. The same applies to isolates under numbers 12 and 13 in Table 2.
In Figure 9 f. the caption only partially corresponds to the image. There are conidia, but no ascospores as such. It would be desirable to present photos of bags with spores, not just spores alone. Moreover, in Aspergillus, the teleomorph is often from the genus Eurotium.
Author Response
Reviewer#2:
Comment: It is advisable to indicate the authors of Host plants (lines 68-71) at the first mention of the genus and species of the plant in accordance with the International Code of Nomenclature for algae, fungi, and plants. The same applies to isolates under numbers 12 and 13 in Table 2
Response: Thanks for the comments. We have corrected both places in the text.
Comment: In Figure 9 f. the caption only partially corresponds to the image. There are conidia, but no ascospores as such. It would be desirable to present photos of bags with spores, not just spores alone. Moreover, in Aspergillus, the teleomorph is often from the genus Eurotium.
Response: Figure 9 is replaced by new photos. The photos of conidia, ascomata, and ascospores are there now.
Reviewer 3 Report
The authors isolated 85 endophytic fungi from mangroves in order to evaluate the production of tannases. Only thirteen of them produced tannases and finally, enzymatic characterization tests were performed on two of them. The article presents information on the genera of tannase-producing fungi, however, the manuscript presents spelling errors and poor fungal terminology. The figures should be improved. The discussion is little and does not focus on the genera found, the importance of tannases for these fungi or their possible contrast in production with other authors. For all these reasons I believe that the article should be rejected. I add comments so that it can be revised and can be evaluated in the future in another journal.
Please check the comments

Author Response
Reviewer#3:
Abstract
Comment: Please check the data because in this section it is mentioned that 84 isolates were evaluated, while in the results section it is mentioned 85
Response: Many thanks. A total of 85 isolates were obtained during isolation. It is a typographical error. Corrected.
Material and methods
Comment:: Line 84.- “1% sodium hypochlorite (NaOCl) and 75% ethanol solution was used for surface sterilization”. Please add the washing times of both substances.
Response: Surface sterilization followed the standard protocol, as referenced in the text. Guo et al. (2000). In 1% sodium hypochlorite (NaOCl) and 75% ethanol solution, the washing time was 1 min.
Comment: Line 133. “The specific activity of an enzyme was calculated by dividing the enzyme activity by protein concentration.” Please change the sentence color from red to black
Response: Corrected
Results.
Comment: Line 206.- “In this study eleven different genera were encountered from different mangrove plants 206 including demataceous fungi, most of isolates were belong to ascomycetes and anamorphic ascomycetes (Deuteromycetes)” Please remove “Deuteromycetes” as it is a term in obsolete in modern mycology. In this sentence it is said that there were eleven genera and in the results only ten genera appear, please revise the writing.
Response: Thanks for pointing out the errors. We have removed the term Deuteromycetes and corrected the error in several genera encountered during the study.
Comment: Line 207.- “Basidiomycetus isolates were very few” Please explain what you are referring to in this sentence, since in the results section no genera belonging to this phylum are described nor in the isolates that the authors identified molecularly
Response: Thanks for the comment. I want to mention here that 85 pure isolates were obtained from selected plants during the isolation of endophytic fungi. Some isolates were unable to produce reproductive structures even after very long incubation. A clamp connection was seen in the mycelium to identify morphologically. Therefore, it was recognized as a basidiomycete. Hence, we have concluded in the text that basidiomycetes isolates also colonize the mangrove plants.
Comment: Line 214.- Please change “Phyllostricta” by “Phyllosticta”
Response: Corrected
Comment: Line 348.- Please change “Phyllostricta” by ‘Phyllosticta’
Response: Corrected
Discussion:
Comment: Line 377. “In the present study, it was found that mangrove plants were predominantly colonized by Ascomycetous fungi and most dominat genus was Chaetomium and Microascus, isolated from all selected mangrove plants”. There is no mention in the results section to support that there were isolates identified within the genus Microascus. please explain this part better and if necessary add it in the results section.
Response: Thanks for the comments and the suggestions. Overviews of all fungal genera obtained are included in the results section. In this study, we mentioned only tannase-producing isolates in the manuscript. The suggestions are included in the revised manuscript.
Comment: Line 392.- The fungal genus Phyllosticta, known as a plant pathogen, causes leaf spot disease isolated from the healthy leaf of Avicennia marina, which is first reported as an endophyte. This sentence is not true since Phyllosticta has been found as an endophyte in leaves of other plants.
Response: Many thanks for the correction. This is the first report of tannase-producing mangrove-associated Phyllosticta.
Comment: Line 402.- “In the present study, a total of 13 isolates were obtained as potent tannase enzyme pro- 402 ducers (Table 1 & 2). The most tannase-producing genera were identified as Chaetomium, 403 Aspergillus, Curvularia, Alternaria, Corynespora, and Phyllostrica”
Please discuss better this section since different genera of fungi were isolated, it would be important to mention if there are works reported for each of the genera including species. It
would also be important to say what advantages it has for an endophyte/saprophyte to have these enzymes
Response: Thanks for the suggestion. We have included all tannase-producing isolates in the discussion part along with their ability. After a detailed review, it was found that most mangrove endophytic isolates, such as Alternaria, Curvularia, and Corynespora, have not been reported as tannase producers.
Comment: Figure 3.- Image should be improved
Response: Corrected.
Comment: Figure 8.- Please change “gm” to “mg”
Response: Corrected.
Round 2
Reviewer 1 Report
Thank You for Your revised MS.
/
Reviewer 2 Report
The authors took into account the comments and suggestions. The article is worthy of publication in a high- ranking journal. It meets the requirements and contains valuable scientific material.
Nj applicable.
Reviewer 3 Report
The current manuscript has been modified with all requested grammatical corrections. The English was thoroughly revised and improved. The discussion was improved. For all the above reasons the article is eligible for publication in this journal.
Comments:
Grammatical style corrections were made.
Improvement of the quality of the figures were made.
English was greatly improved.
For all of the above, this article is now ready for publication